# Genome-Wide Identification of PAP1 Direct Targets in Regulating Seed Anthocyanin Biosynthesis in *Arabidopsis*

**DOI:** 10.3390/ijms242216049

**Published:** 2023-11-07

**Authors:** Yuan Guo, Dong Li, Tiantian Liu, Yuxin Li, Jiajia Liu, Mingyuan He, Xiaohui Cui, Zijin Liu, Mingxun Chen

**Affiliations:** 1Shaanxi Key Laboratory of Crop Heterosis, National Yangling Agricultural Biotechnology and Breeding Center, College of Agronomy, Northwest A&F University, Yangling 712100, China; guoyuan2109@163.com (Y.G.); ltt786@163.com (T.L.); l18821673700@163.com (Y.L.); liu18235568734@163.com (J.L.); outf4@foxmail.com (M.H.); 18248047704@163.com (X.C.); liuzijin@nwafu.edu.cn (Z.L.); 2The Engineering Research Institute of Agriculture and Forestry, Ludong University, Yantai 264025, China; 3128@ldu.edu.cn

**Keywords:** PAP1, MBW complex, anthocyanin biosynthesis, seeds, *Arabidopsis*

## Abstract

Anthocyanins are widespread water-soluble pigments in the plant kingdom. Anthocyanin accumulation is activated by the MYB-bHLH-WD40 (MBW) protein complex. In *Arabidopsis*, the R2R3-MYB transcription factor PAP1 activates anthocyanin biosynthesis. While prior research primarily focused on seedlings, seeds received limited attention. This study explores PAP1’s genome-wide target genes in anthocyanin biosynthesis in seeds. Our findings confirm that *PAP1* is a positive regulator of anthocyanin biosynthesis in *Arabidopsis* seeds. PAP1 significantly increased anthocyanin content in developing and mature seeds in *Arabidopsis*. Transcriptome analysis at 12 days after pollination reveals the upregulation of numerous genes involved in anthocyanin accumulation in *35S:PAP1* developing seeds. Chromatin immunoprecipitation and dual luciferase reporter assays demonstrate PAP1’s direct promotion of ten key genes and indirect upregulation of *TT8*, *TTG1,* and eight key genes during seed maturation, thus enhancing seed anthocyanin accumulation. These findings enhance our understanding of PAP1’s novel role in regulating anthocyanin accumulation in *Arabidopsis* seeds.

## 1. Introduction

Anthocyanins, water-soluble pigments, fall under the flavonoids class of secondary metabolites, contributing red, purple, and blue hues to various fruits and vegetables [1]. These pigments play a role in attracting pollinators and seed dispersers agents [1,2]. Anthocyanin formation is influenced by environmental factors such as UV irradiation [3], temperature [4], drought [5], and nutrient deficiency [6]. Under adverse conditions, anthocyanin concentrations generally increase, indicating their involvement in biotic and abiotic stress responses [7,8,9]. Anthocyanins, with their antioxidant properties, also serve as vital micronutrients for humans, guarding against cardiovascular, neurodegenerative, metabolic diseases, and cancer [10]. Therefore, gaining a deeper understanding of anthocyanin accumulation and its regulatory mechanisms holds significant scientific and economic importance.

The anthocyanin biosynthetic pathway is a major branch of the general phenylpropanoid pathway that starts with phenylalanine (Phe) [11]. This pathway can be briefly divided into three parts: beginning steps of the general phenylpropanoid pathway, early steps of the flavonoid pathway, and late steps of the anthocyanin specific pathway. The expression of structural genes in the anthocyanin biosynthetic pathway primarily depends on the MYB-bHLH-WD40 (MBW) transcription complex, comprising MYB and basic helix-loop-helix (bHLH) transcription factors alongside WD40 proteins [12,13]. MYB proteins, one of the largest plant transcription factor families, play roles in cell differentiation [14], stress responses [15], metabolism [16], and development processes [17]. R2R3-MYB transcription factors, featuring an N-terminal conserved MYB domain and a C-terminal variable activation or repression domain, play a key role in determining the spatial and temporal patterns of anthocyanin accumulation [16,18]. *ZmC1* (*colorless1*), the first discovered R2R3-MYB transcription factor, was reportedly essential for anthocyanin biosynthesis in maize aleurone tissues [19]. Several R2R3-MYB members act as positive regulators for anthocyanin biosynthetic genes, including *PRODUCTION OF ANTHOCYANIN PIGMENTATION 1* (*PAP1*)/*MYB75*, *PAP2/MYB90*, *MYB113*, and *MYB114* in *Arabidopsis* [20,21], *MdMYB10* and *MdMYB110a* in apple [22], and *MYB78* in canola [23]. Conversely, specific R2R3-MYB transcription factors repress anthocyanin biosynthesis [24,25], including *MYB3*, *MYB4*, and *MYB6* in *Arabidopsis* [26] and *MdMYB16*, *MdMYB17*, and *MdMYB111* in apple [27]. 

*PAP1*, also known as *MYB75*, functions as a key anthocyanin biosynthesis regulator [28,29]. In *Arabidopsis* seedlings, the *pap1* loss-of-function mutant or *PAP1* RNA interference plants displayed reduced anthocyanin induction [21,30]. Conversely, over-expression of *PAP1* in the activation-tagged *PAP1* gain-of-function mutant (*pap1-D*) resulted in hyperaccumulation of anthocyanins in leaves, roots, stems, and flowers [20]. For governing the anthocyanin biosynthetic pathway, PAP1 forms a complex with bHLH anthocyanin regulators GLABRA 3, ENHANCER OF GLABRA 3, and TRANSPARENT TESTA 8 (TT8), along with a WD40-repeat protein, TRANSPARENT TESTA GLABRA 1 (TTG1). This complex enhances the expression of late anthocyanin biosynthetic genes: *dihydroflavonol-4-reducatse* (*DFR*), *leucoanthocyanidin dioxygenase* (*LDOX*), and *UDP-glucose: flavonoid-3-O-glucosyl-transferase* (*UF3GT*) [11,21,28]. Further studies have demonstrated that the post-translational modification of MBW proteins modulates the MBW protein complex’s transcriptional activity. PAP1 degradation in the dark is mediated by the CONSTITUTIVE PHOTOMORPHOGENIC 1/SUPPRESSOR of PHYA-105 ubiquitin ligase [31], while phosphorylation by MAP KINASE 4 stabilizes PAP1, which is essential for light-induced anthocyanin accumulation [32]. Another post-translational modification hinders the formation of the MBW protein complex. The DNA-binding homeodomain ZIP transcription factor HAT1 interferes with the formation of the MBW protein complex by interacting with PAP1 and recruiting the TOPLESS corepressor to epigenetically modulate anthocyanin biosynthetic genes [33]. Furthermore, the phosphate starvation signaling pathway repressor SPX4 physically interacts with PAP1, disrupting the PAP1–TT8 interaction and impairing the transcriptional activation of the anthocyanin biosynthesis gene *DFR* [34]. A recent study reveals that PHYTOCHROME-INTERACTING Factor 4 competes with TT8 to bind PAP1, thus affecting the regulation of the MBW protein complex in anthocyanin biosynthesis [35].

These findings suggest that the MBW complex serves as a central regulatory hub for anthocyanin accumulation. Notably, PAP1 is a key activator in the anthocyanin biosynthesis pathway with conserved functions in various crops [11,21,28]. However, genome-wide targets of PAP1 in *Arabidopsis* seeds remain unexplored. In our study, we demonstrated that *PAP1* enhances seed anthocyanin accumulation by upregulating some anthocyanin biosynthesis-related genes during *Arabidopsis* seed development. Our results offer new insights into PAP1’s regulatory role in *Arabidopsis* seed anthocyanin accumulation.

## 2. Results

### 2.1. Positive Correlation of PAP1 Levels and Anthocyanin Accumulation in Seeds

To validate PAP1’s role in anthocyanin accumulation in *Arabidopsis* seeds, we introduced the *35S:PAP1-6HA* construct into *Arabidopsis* Columbia-0 (Col-0), yielding fifteen transgenic lines. We selected three independent *35S:PAP1* T_3_ homozygous transgenic lines with the highest PAP1 expression levels, *35S:PAP1 #1*, *35S:PAP1 #3*, and *35S:PAP1 #5*, for subsequent experiments (Figure 1).

Compared to wild-type plants, the developing seeds of these three *PAP1*-over-expressing transgenic lines exhibited enhanced pigmentation at 10 and 12 days after pollination (DAP) as well as in mature seeds (Figure 2A–C). Additionally, the seedlings of the transgenic lines (*35S:PAP1 #1*, *35S:PAP1 #3*, and *35S:PAP1 #5*) displayed purple stems and petioles, while wild-type seedlings were green (Appendix A).

Furthermore, we measured the anthocyanin levels in mature seeds of the wild-type and three transgenic lines, revealing a significant increase in anthocyanin content in the transgenic lines compared to wild-type plants (Figure 2D). Proanthocyanidins (PAs) are a class of oligomeric or polymeric flavonoids. The intermediate compound dihydroflavonol can be further converted to anthocyanins or PAs through distinct branches of the flavonoid pathway [36]. Previous studies have shown that PAs accumulate in the seed coat and protect the embryo and endosperm [37]. Consequently, we assessed the PAs levels in mature seeds of both the wild-type and the three transgenic lines, revealing no significant difference in PAs content (Figure 2E). In summary, our findings suggested that PAP1 selectively regulates anthocyanin accumulation, but not PAs, in seeds during *Arabidopsis* seed development.

### 2.2. A Whole-Genome Analysis of Genes Associated with Seed Anthocyanin Accumulation

To elucidate the regulatory mechanism of *PAP1* in seed anthocyanin accumulation, we performed RNA-Sequencing (RNA-Seq) analysis on developing seeds from the transgenic line *35S:PAP1 #5* and wild-type Col-0 plants at 12 DAP. The results identified 5174 differentially expressed genes (DEGs), with 4760 upregulated and 414 downregulated DEGs (Table 1). Among them, seventy-four upregulated genes (1.6%) and three downregulated genes (0.7%) were involved in flavonoid biosynthesis (Table 1). Additionally, 30% of upregulated genes participated in primary metabolic processes, including carbohydrate metabolism (12.8%), nucleic acid (5.2%), amino acid and protein (5.0%), cell wall (3.3%), and photosynthesis (2.3%) (Table 1). Notably, 780 upregulated genes (16.4%) and 95 downregulated genes (22.9%) were linked to stress/defense responses (Table 1). These findings highlight PAP1’s pivotal role in seed anthocyanin accumulation and other crucial physiological and biochemical processes.

### 2.3. Validation of Seed Anthocyanin Accumulation-Related Genes

To confirm the regulation of anthocyanin biosynthesis-related genes in developing *35S:PAP1 #5* seeds at 12 DAP and to identify target genes controlled by PAP1 in the seed anthocyanin biosynthetic pathway, we conducted quantitative real-time PCR (RT-qPCR) to compare expression patterns between *35S:PAP1 #5* and wild-type plants. We selected twenty highly upregulated genes, including two transcription factors (*TT8* and *TTG1*) and eighteen structural genes (Table 2). The RT-qPCR results aligned with the RNA-seq data, confirming significant upregulation of these twenty genes (Figure 3 and Table 2). Notably, pivotal genes involved in Phe synthesis, such as arogenate dehydratase 5 (*ADT5*), and genes related to Phe metabolic pathways, such as cinnamate 4-hydroxylase (*C4H*) and 4-coumarate: CoA ligase 3 (*4CL3*), exhibited higher expression in *35S:PAP1 #5* compared to the wild-type seeds at 12 DAP (Figure 3). Additionally, genes associated with anthocyanin biosynthesis, like chalcone synthase (*CHS)*, chalcone isomerase (*CHI*), flavanone 3-hydroxylase (*F3H*), flavonoid 3′-hydroxylase (*F3’H*), *DFR*, and anthocyanidin synthase (*ANS*), as well as genes involved in anthocyanin modification and transport, including flavonoid 3-*O*-glycosyltransferase (*3GT*), anthocyanin 5-*O*-glycosyltransferase (*5GT*), *UF3GT*, UDP-glycosyltransferases (*UGT79B2* and *UGT79B3*), anthocyanin 3-*O*-glucoside-6″-*O*-acyltransferases (*3AT1* and *3AT2*), anthocyanidin 5-*O*-glucoside-6″-*O*-malonyltransferase (*5MAT*), and glutathione *S*-transferase 26 (*GST26*), were likewise upregulated in *35S:PAP1 #5* at 12 DAP (Figure 3). In summary, these findings underscore the role of PAP1 in enhancing seed anthocyanin accumulation by activating the expression of regulatory and structural genes involved in anthocyanin biosynthesis, modification, and transport. 

### 2.4. PAP1 Promotes Anthocyanin Accumulation by Directly Activating the Expression of ADT5, CHS, F3H, DFR, ANS, 3GT, UGT79B2, UGT79B3, 5MAT, and GST26 in Arabidopsis Developing Seeds

To investigate PAP1’s regulation of seed anthocyanin accumulation, we performed chromatin immunoprecipitation (ChIP) assays on developing siliques at 12 DAP from *35S:PAP1-6HA #5* plants. This allowed us to understand how PAP1 controls the transcription of target genes. From the twenty genes mentioned earlier, we selected *ADT5*, *CHS*, *F3H*, *DFR*, *ANS*, *3GT*, *UGT79B2*, *UGT79B3*, *5MAT*, and *GST26* due to their possession of PAP1 binding sites. The core binding motif of PAP1, identified as a 7 bp MYB-recognizing element (MRE) (ANCNNCC), was found in the promoter regions of these ten genes [34,63]. We designed primers to cover all possible MRE sites bound by PAP1 in the promoter regions of these genes. There are three MREs within the promoter of *ADT5*, five MREs in *CHS*, three MREs in *F3H*, two MREs in *DFR*, three MREs in *ANS*, three MREs in *3GT*, two MREs in *UGT79B2*, two MREs in *UGT79B3*, three MREs in *5MAT*, and four MREs in *GST26* (Figure 4). The ChIP assay revealed that PAP1-6HA was associated with specific promoter regions: P3 of *ADT5*, P1 of *CHS*, P1 of *F3H*, P1 of *DFR*, P1 and P2 of *ANS*, P1 and P2 of *3GT*, P2 of *UGT79B2*, P1 and P2 of *UGT79B3*, P3 of *5MAT*, and P2 of *GST26* (Figure 4). These results demonstrated that PAP1 directly binds to the promoter regions of *ADT5*, *CHS*, *F3H*, *DFR*, *ANS*, *3GT*, *UGT79B2*, *UGT79B3*, *5MAT*, and *GST26* to promote their expression. 

Moreover, we further evaluated the positively regulatory function of PAP1 on the transcription of *ADT5*, *CHS*, *F3H*, *DFR*, *ANS*, *3GT*, *UGT79B2*, *UGT79B3*, *5MAT*, and *GST26* using a transient dual-luciferase reporter assay. We constructed effectors with or without the CDS of PAP1, and reporters containing firefly luciferase driven by the promoters of *ADT5*, *CHS*, *F3H*, *DFR*, *ANS*, *3GT*, *UGT79B2*, *UGT79B3*, *5MAT*, and *GST26* along with Renilla luciferase driven by the 35S promoter (*Pro35S:REN*) was used as an internal control (Figure 5A). After co-expression with effector and reporter in *N. benthamiana* leaves, the LUC/REN ratios of *ADT5*, *CHS*, *F3H*, *DFR*, *ANS*, *3GT*, *UGT79B2*, *UGT79B3*, *5MAT*, and *GST26* were markedly increased (Figure 5B), indicating that PAP1 directly activates their transcription in *N. benthamiana* leaves. In summary, our findings collectively indicate that PAP1 promotes anthocyanin accumulation by directly activating the expression of *ADT5*, *CHS*, *F3H*, *DFR*, *ANS*, *3GT*, *UGT79B2*, *UGT79B3*, *5MAT*, and *GST26*, while also indirectly promoting the expression of *C4H*, *4CL3*, *CHI*, *F3′H*, *5GT*, *UF3GT*, *3AT1*, *3AT2*, *TT8*, and *TTG1* during seed development in *Arabidopsis.*

## 3. Discussion

Anthocyanin accumulation is a dynamic phenomenon in various plant species. The transcription factor PAP1, an R2R3-MYB member, serves as a pivotal hub, integrating diverse internal and external stimuli affecting anthocyanin biosynthesis, with prior research mainly focused on seedling phenotypes [28,29]. However, information on the direct targets of the PAP1 in the regulation of anthocyanin accumulation, especially in seeds of *Arabidopsis*, remains limited. In this study, we identified new genes targeted by PAP1, directly or indirectly regulating anthocyanin accumulation at the genome-wide level in *Arabidopsis* seeds.

Numerous R2R3-MYB genes are known to positively or negatively regulate anthocyanin biosynthesis [24,64]. Previous reports indicated that two *PAP1* over-expressing lines, the *pap1-D* mutant and the *PAP1* cDNA over-expressing transgenic plant, exhibited similar anthocyanin accumulation in vegetative tissues but differed in seed color and accumulation patterns of anthocyanins and PAs in seeds [20,65]. The *pap1-D* mutant showed increased pigmentation in leaves, stems, and roots but no change in seed color [20,65]. In contrast, the *PAP1* cDNA over-expressing plant displayed darker colors in all vegetative organs, including seeds [65]. Soluble anthocyanin levels increased in seeds of transgenic plants over-expressing *PAP1* cDNA but remained unchanged in seeds of the *pap1-D* mutant [65]. Consistent with results from *PAP1* cDNA over-expressing lines, our reverse genetic approach demonstrated that PAP1 over-expression modulates anthocyanin biosynthesis, leading to anthocyanin hyperaccumulation in both developing and mature seeds of *Arabidopsis* (Figure 2A–C). The increased anthocyanin content should be the reason for the darker color of mature seeds in transgenic lines (Figure 2C,D). Notably, total PAs content showed no difference between the wild-type and transgenic lines in mature seeds (Figure 2D). Successful PAs accumulation in *Arabidopsis* reportedly requires the cooperation of multiple genes [51]. Based on this observation and previous findings showing increased PAs content in the *pap1-D* mutant but decreased PAs content in *PAP1* cDNA over-expressing plants in *Arabidopsis* seeds [65], it is reasonable to suggest that PAP1’s influence on accumulation of PAs is more complex than anthocyanin production in *Arabidopsis* seeds, likely due to various unknown factors. Further research is needed to understand the relationships between PAP1 and PAs in *Arabidopsis* seeds. Therefore, we speculate that PAP1 plays a specific positive role in anthocyanin accumulation, not PAs, during seed development in *Arabidopsis* seeds. 

The regulation of gene expression involved in the anthocyanin biosynthetic pathway is largely coordinated by a complex network of interactions between transcription factors and their target genes [21,34]. The transcriptome analysis revealed that seventy-four upregulated genes (1.6%) and three downregulated genes (0.7%) in developing seeds of *35S:PAP1 #5* were related to flavonoid metabolism (Table 1 and Appendix A). Additionally, a significant portion of all DEGs in developing seeds of *35S:PAP1 #5* (16.9%) were associated with stress/defense responses (Table 1), consistent with PAP1’s known role as a stress regulator [66,67]. Previous studies have discovered that anthocyanins play a crucial role in enhancing tolerance to biotic and abiotic stresses in vegetative tissues [8,9]. It is possible that PAP1 directly altered the expression patterns of these differentially expressed stress/defense-responsive genes (Appendix A). Alternatively, the hyperaccumulation of anthocyanins in developing seeds may have indirectly regulated these stress/defense-responsive genes.

Multiple studies confirm PAP1’s potent impact on anthocyanin accumulation in *Arabidopsis* seedlings by specifically inducing anthocyanin-related gene expression. Tohge et al. [28], via microarray experiments showed upregulation of late anthocyanin biosynthetic genes, including glycosyltransferase genes *UGT79B1* (*At5g54060*), *UGT75C1* (*At4g14090*), and *UGT78D2* (*At5g17050*), as well as early genes like *4CL*, *CHS*, *CHI*, and *F3′H* in PAP1 over-expressing plants leaves. RNA gel blot analysis showed massive enhancement of the expression of genes across the entire phenylpropanoid pathway, including *PAL1*, *CHS*, *DFR*, and *GST* in 6-week-old *pap1-D* plants [20]. Our data corroborates this trend, as it shows a substantial upregulation of key genes throughout the entire anthocyanin biosynthetic pathway in developing seeds of *35S:PAP1 #5* (Figure 3). Additionally, molecular analyses reveal PAP1’s direct binding to the promoters of ten structural genes, including *ADT5*, *CHS*, *F3H*, *DFR*, *ANS*, *3GT*, *UGT79B2*, *UGT79B3*, *5MAT*, and *GST26*, activating their transcription during anthocyanin biosynthesis in *Arabidopsis* seeds (Figure 4, Figure 5 and Figure 6). Notably, ADT plays a crucial role in converting arogenate into Phe during sucrose-induced anthocyanin biosynthesis in *Arabidopsis*. There are six ADT isoforms, ADT1 to ADT6, which redundantly regulate anthocyanin biosynthesis with varying contributions. Compared to other *ADTs*, when *ADT4* or *ADT5* were overexpressed, it results in Phe hyperaccumulation and a significant increase in anthocyanin content in *Arabidopsis* seedlings [38]. CHS catalyzes the initial step of the flavonoid pathway, condensing *p*-coumaroyl-CoA and malonyl-CoAs to produce tetrahydroxychalcone [42]. Over-expressing *CHS* enhances high light resistance by increasing anthocyanin synthesis [43]. Foliar application of *CHS*-specific dsRNAs and siRNAs resulted in an efficient downregulation of *CHS* and suppressed anthocyanin accumulation in *Arabidopsis* under anthocyanin biosynthesis-modulating conditions [44]. F3H hydroxylates flavanone to yield dihydrokaempferol [47]. Downregulating *F3H* in strawberries markedly reduces anthocyanin and moderately decrease flavonol content [68]. DFR reduces dihydroflavonol to leucoanthocyanidin, the initial reaction that leads to anthocyanin and proanthocyanidin biosynthesis. Pi starvation was found to destabilize the SPX4-PAP1 complex, allowing PAP1 to directly bind the DFR promoter, activating anthocyanin biosynthesis [34]. ANS, a pivotal enzyme in anthocyanin biosynthetic, converts colorless leucoanthocyanins into colored anthocyanidins. *Arabidopsis ans* mutants display reduced anthocyanin accumulation in hypocotyls [51] and rosette leaves [69]. 3GT, identified as flavonoid 3-*O*-glucosyltransferase, was predicted to participate in *Arabidopsis*’ anthocyanin biosynthesis [28]. UDP-glycosyltransferases (UGTs) catalyze the final step in anthocyanin biosynthesis, resulting in diverse anthocyanin molecules in *Arabidopsis* [11]. Two UGTs, UGT79B2 and UGT79B3, modified anthocyanins by adding UDP-rhamnose to cyanidin and 3-*O*-glucoside-cyanidin. Ectopic expression of *UGT79B2/B3* significantly boosts anthocyanin levels, but *ugt79b2/b3* double mutants exhibited reduced anthocyanin levels [5]. 5MAT encodes malonyl-CoA cyanidin 3,5-diglucoside transferase activity, thereby accelerating malonylated anthocyanin accumulation [57]. GST26, encoding an *Arabidopsis* glutathione *S*-transferase-like protein, is implicated in anthocyanin transport into vacuoles [58]. 

However, we observed that PAP1 indirectly regulated eight upregulated structural genes essential for anthocyanin biosynthesis in *Arabidopsis* seeds. These genes are *C4H*, *4CL3*, *CHI*, *F3′H*, *5GT*, *UF3GT*, *3AT1*, and *3AT2*. C4H and 4CL are the second and third enzymes in the phenylpropanoid pathway, converting Phe to *p*-coumaroyl CoA [39,40,41]. CHI uses tetrahydroxychalcone to produce naringenin, and its mutation fails to accumulate anthocyanins [45,46]. F3′H hydroxylates dihydrokaempferol into dihydroquercetin [48,49]. 5GT encodes anthocyanin 5-*O*-glucosyltransferase, and UF3GT converts cyanidin 3-*O*-glucoside to cyanidin 3-*O*-xylosyl glucoside [28,55]. 3AT1 and 3AT2 encode coumaroyl CoA-cyanidin 3-*O*-glucose transferase and have redundant functions in anthocyanin stability [56].

Furthermore, we demonstrated that PAP1 indirectly enhanced the expression of two regulatory genes, *TT8* and *TTG1*, during seed anthocyanin biosynthesis (Figure 3). This activation is instrumental in accelerating anthocyanin production in seeds (Figure 2). The feedback mechanisms between the MYB and bHLH components of the MBW activation complex play a pivotal role in flavonoid regulation. TT8, a bHLH transcription factor, regulates its own expression via a positive feedback loop through an MBW complex, contributing to anthocyanin and PA biosynthesis regulation [60,70]. TTG1, encoding a WD40 repeat transcription factor, collaborates with TT8 and TT2 (encoding MYB123) to mediate anthocyanin pigment production in developing seeds [59]. Our data suggests that PAP1, an R2R3-MYB factor, upregulates *TT8* and *TTG1* expression within the MBW complex for anthocyanin gene regulation, with the hierarchical regulation of TT8 and TTG1 expression requiring further investigation. 

*PAP1* expression has the potential to significantly enhance anthocyanin accumulation in many plant species, resulting in a dark purple color in various plant organs [20,21,65]. It has been demonstrated that the increased anthocyanin accumulation confers plants with enhanced tolerance to abiotic stress [66,67]. Thus, there is an interesting question needing to be investigated: do the enhanced anthocyanin levels in seeds enhance abiotic stress tolerance? Previous studies indicated that the anthocyanins present in seed extracts could serve as defense molecules against abiotic stresses such as UVB radiation, drought, and low or high temperatures [71]. Recently, some researchers have observed an association between seed dormancy and seed color [72,73]. These findings suggested that PAP1 may be a valuable candidate gene that is associated with seed dormancy and germination under stress due to its increased anthocyanin content.

In summary, our study reveals that the R2R3-MYB transcription factor PAP1 directly activates the expression of ten anthocyanin biosynthetic pathway-related structural genes, *ADT5*, *CHS*, *F3H*, *DFR*, *ANS*, *3GT*, *UGT79B2*, *UGT79B3*, *5MAT*, and *GST26*, during seed development (Figure 6). This makes PAP1 a promising target for genetic manipulation to enhance seed anthocyanin levels, improving seed anthocyanin quality.

## 4. Materials and Methods

### 4.1. Plant Materials and Growth Conditions

The *Arabidopsis thaliana* ecotype Col-0 served as the wild-type control. As previously stated [74], the plants were grown in a 16 h light/8 h dark cycle at 22 °C in a growth chamber with overhead light at 160 μmol m^−2^ s^−1^.

### 4.2. Plasmid Construction and Transgenic Plants Generation

Specific primers were designed from the full-length coding sequence (CDS) of *PAP1* (*At1g56650*) without a stop codon from the TAIR database. To create *35S:PAP1-6HA*, the CDS of *PAP1* (without the stop codon) was amplified using PAP1_F and PAP1_R primers (Appendix A). PCR products were digested using XmaI and SpeI, then ligated into the pGreen-35S-6HA binary vector to create an in-frame fusion of PAP1-6HA under the 35S promoter. 

The *35S:PAP1-6HA* vector was transformed into *Arabidopsis* Col-0 using the *Agrobacterium-tumefaciens*-mediated floral dip method [75]. *35S:PAP1* transgenic plants were selected based on their soil using Basta, and successful transformation was confirmed through DNA genotyping until homozygous T_3_ transgenic progenies were obtained.

### 4.3. Phenotypic Observation of Seeds Color and Seed Size

To analyze seed color and seed size, the immature (10 and 12 DAP) and mature seeds of three *Arabidopsis* transgenic lines were randomly selected from major inflorescences and photographed using a SZ61 stereomicroscope (Olympus, Tokyo, Japan).

### 4.4. Determination of Anthocyanin and PAs Content

Anthocyanin measurement followed the protocol by Li et al. [76] with minor adjustments. Seeds were briefly frozen in liquid nitrogen, ground in a mortar, and approximately 5 mg of seed powder was placed in a 10 mL graduated test tube and incubated overnight at 4 °C in 3 mL methanol solution with 1% (*v*/*v*) HCl. After a 60 min incubation at 75 °C and cooling to room temperature, the sample was centrifuged for 15 min at 1500 rpm (HC-3018R, Zonkia, Anhui, China). The supernatant was mixed with 2 mL of distilled water and an equal volume of chloroform, followed by centrifugation (1500 rpm, 15 min, HC-3018R, Zonkia, Anhui, China). The supernatant’s absorbance was measured at 535 nm using a spectrophotometer (V-1200, Mapada, Shanghai, China) and the anthocyanin content was then normalized to the dry seed weight.

PAs extraction, adapted from Kitamura et al. [77], involved grinding mature seeds (10 mg) and mixing the powder with 1.5 mL of 70% (*v*/*v*) acetone extraction buffer containing 5.26 mM Na_2_S_2_O_5_. This mixture was sonicated using an ultrasonic bath (SB-5200 DT, Scientz, Ningbo, China) for 20 min at room temperature and then centrifuged at 1500 rpm for 15 min (HC-3018R, Zonkia, Anhui, China). The supernatant was dried and resuspended in HCl:butanol:70% acetone (2:10:3). The resulting absorbance was measured at 545 nm using an Infinite M200 PRO (Tecan). Following this, the solution was heated at 95 °C for 60 min and the absorbance at 545 nm was recorded again. The soluble PA fraction was calculated by subtracting the initial absorbance from the final one. The pellet obtained after 70% acetone extraction was dried via evaporation, resuspended in the HCl/butanol solution, and hydrolyzed to determine the insoluble PAs. Three independent biological replicates were conducted, each with three technical repetitions.

### 4.5. RNA-Seq Analysis

The RNA-seq experiment utilized developing seeds at 12 DAP from the major inflorescences of wild-type (Col-0) and *Col-0 35S:PAP1 #5* over-expressing plants. Three independent biological replicates from each genotype were sequenced using BGI-Tech (Shenzhen, China), following the standard protocol (http://bgitechsolutions.com/sequencing/45 (accessed on 18 November 2019)). DEGs were identified using |log_2_ ratios| ≥ 1 and a *false discovery rate* (FDR) of ≤0.05 (Appendix A). DEGs underwent functional classification based on the biological process category of Arabidopsis Gene Ontology (http://www.geneontology.com (accessed on 6 February 2020)). 

### 4.6. RNA Extraction and RT-qPCR Analysis

Total RNA extraction from 12 DAP developing seeds was performed using the SteadyPure Plant RNA Extraction Kit (Accurate Biology, Changsha, China), followed by cDNA reverse transcription (TransGen, Beijing, China). RT-qPCR analysis was conducted with three independent biological replicates using SYBR Green Master Mix (Cofitt, Hongkong, China) on the QuantStudio^TM^ 7 Flex Real-Time System (Thermo Fisher Scientific, Waltham, MA, USA). The *Arabidopsis* house-keeping gene *EF1αA4* served as the internal control, and relative expression values of the target genes were calculated via normalization against *EF1αA4* using a modified double-delta method [78]. The RT-qPCR primer details are listed in Appendix A.

### 4.7. ChIP-qPCR Assay

The ChIP-qPCR assay followed a previously described protocol [79]. Developing siliques (3−5 g) at 12 DAP were harvested from both wild-type (Col-0) and *Col-0 35S:PAP1 #5* over-expressing plants. The samples underwent triple ddH_2_O washes and were cross-linked using 1% (*v*/*v*) formaldehyde (37 mL) under vacuum on ice for 15 min. Crosslinking was terminated by adding 2.5 mL of 2 M glycine. After being ground in liquid nitrogen, nuclear protein was separately extracted using sucrose-based buffers containing 0.4, 0.25, and 1.7 M sucrose. Chromatins were isolated, and DNA was sheared into 200–700 bp fragments through sonication with ultrasonic cell disruptors (Scientz-IID, Scientz, Ningbo, China). After centrifugation at 4 °C for 5 min at 12,000 rpm (Sorval Legend^TM^ Micro 17, Thermo Fisher Scientific, Waltham, MA, USA), the chromatin remained in the upper aqueous phase. PAP1-6HA chromatin DNA was immunoprecipitated overnight using anti-HA magnetic beads (Thermo, USA) at 4 °C. The beads were washed and collected using a magnetic rack, and the immune complexes were eluted twice. Subsequently, the complexes were eluted and reversely crosslinked at 65 °C for 10 h in 5 M NaCl. Proteins were digested with 0.5 M EDTA, 1 M Tris-HCl (pH 6.5), and 3 mL of proteinase K (10 mg/mL) at 45 °C for 1 h. The DNA fragments were extracted using a Phenol/chloroform/isoamyl alcohol solution (25:24:1, pH > 7) and stored at −80 °C. The relative enrichment of each fragment was assessed via RT-qPCR. Each experiment involved three biological replicates with three technical replicates per biological replicate. *Arabidopsis EF1αA4* and *ACTIN7* served as the internal reference and negative control, respectively. The ChIP-qPCR assay primer details are provided in Appendix A.

### 4.8. Transient Dual-Luciferase Reporter Analysis

The *PAP1* CDS was amplified and cloned into pGreenII 62-SK under the 35S promoter to form effector constructs. The effector constructs without *PAP1* served as the empty control. Promoters for *ADT5*, *CHS*, *F3H*, *DFR*, *ANS*, *3GT*, *UGT79B2*, *UGT79B3*, *5MAT*, and *GST26* were separately cloned into pGreenII 0800-LUC [80] to form reporter constructs. *A. tumefaciens* strain GV3101 was transformed with all the constructs along with pSoup-P19 (Weidi Biotechnology, Shanghai, China). Effector and reporter constructs were mixed in a buffer comprising 10 mM MgCl_2_, 10 mM MES-KOH (pH 5.8), and 10 μM acetosyringone in a 1:1 ratio and then injected into the young leaves of 4-week-old *N. benthamiana*. The infiltrated plants were cultured in a climate incubator (RXD-1000D-LED, Prandt, Ningbo, China) with a light/dark 16:8 h photoperiod cycle at 22 °C for 72 h. The firefly luciferase (LUC) and Renilla luciferase (*REN*, an internal control) activities were assessed using a dual-luciferase reporter assay kit (YEASEN, Shanghai, China) on a multifunctional enzyme label instrument (Spark^®^, Tecan, Männedorf, Switzerland). Six independent biological samples were examined. The primers for the dual-luc assay are provided in Appendix A.

### 4.9. Statistical Analysis

This study used a completely randomized design. Data were expressed as mean and standard deviation and analyzed using one-way analysis of variance (ANOVA) via SPSS software (version 17.0, SPSS Inc., Chicago, IL, USA). Significant differences were determined using a two-tailed paired Student’s *t*-test at the 0.05 significance level.

## Figures and Tables

**Figure 1 ijms-24-16049-f001:**
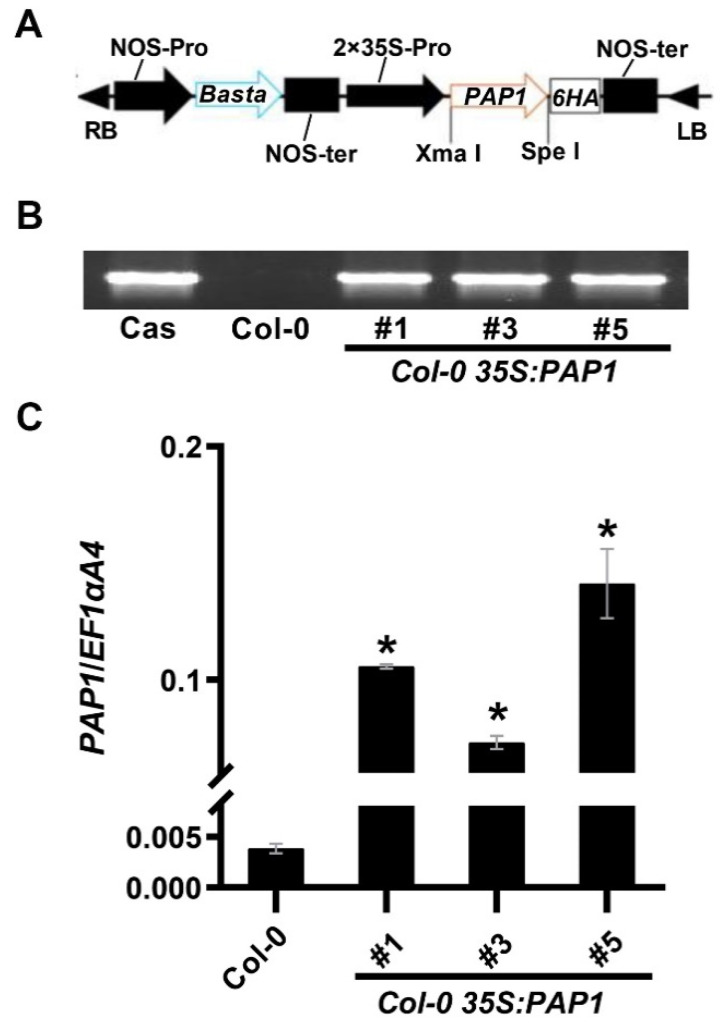
Characterization of *Col-0 35S:PAP1* lines: (**A**) Schematic diagram of the constitutive expression cassette of the *PAP1* gene in the binary vector pGreen-35S-6HA used for plant transformation. RB, right border; LB, left border; NOS-pro, nopaline synthase promoter; NOS-ter, nopaline synthase terminator; Basta, glyphosate; 35S-pro, CaMV 35S promoter. (**B**) PCR-based DNA genotyping of *Col-0 35S:PAP1* transgenic plants using specific primers for the 35S_P/PAP1_R. Cas, cassette. (**C**) Expression analysis of *PAP1* in wild-type (Col-0) and three independent *Col-0 35S:PAP1* transgenic plants using RT-qPCR. RNA samples were extracted from rosette leaves. Results were normalized against the expression of *Arabidopsis EF1aA4* as the internal control. Values are means ± SD (*n* = 3). Asterisks (*) indicate a significant difference in gene expression in the transgenic plants of *PAP1* compared with Col-0 plants (two-tailed paired Student’s *t*-test, *p* ≤ 0.05).

**Figure 2 ijms-24-16049-f002:**
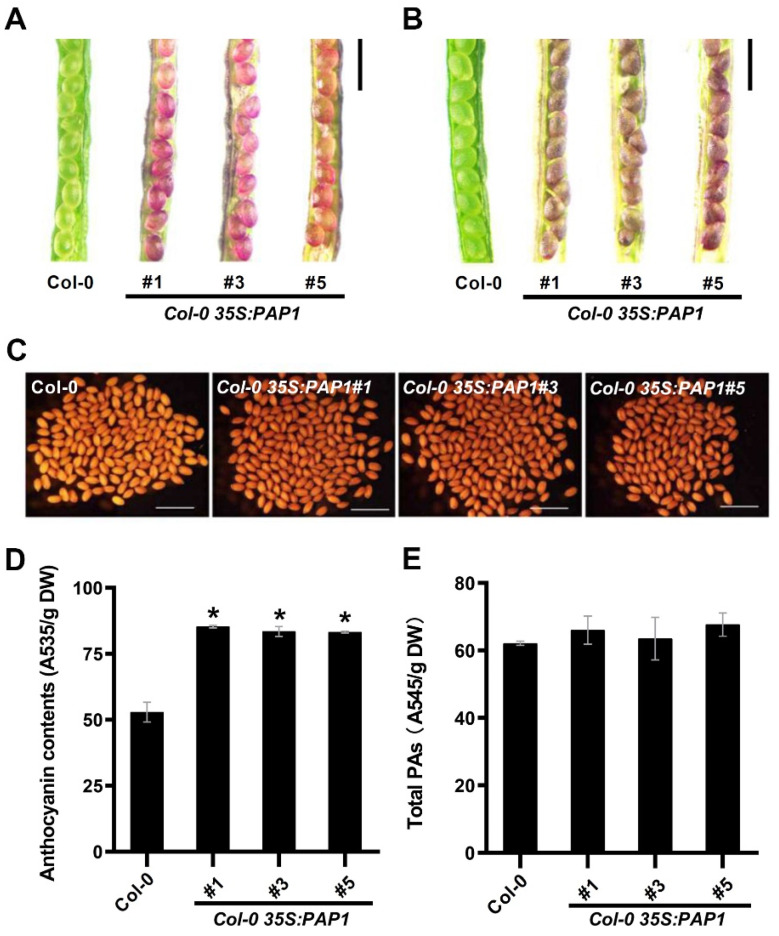
Effect of PAP1 on the accumulation of anthocyanin and PAs in seeds: (**A**–**C**) Phenotypes of wild type (Col-0) and *Col-0 35S:AtPAP1* immature seeds at 10 DAP (**A**), 12 DAP (**B**), and mature seeds (**C**). Scale bars = 1 mm. (**D**) Total anthocyanin contents in mature seeds of wild-type (Col-0) and *Col-0 35S:PAP1*. Asterisks (*) denote the statistically significant differences between the indicated samples (Student’s *t*-test, *p* ≤ 0.05). Values are means ± SD (*n* = 3). DW, dry weight. (**E**) Total PAs contents in mature seeds of wild-type (Col-0) and *Col-0 35S:PAP1*. Values are means ± SD (*n* = 3). DW, dry weight.

**Figure 3 ijms-24-16049-f003:**
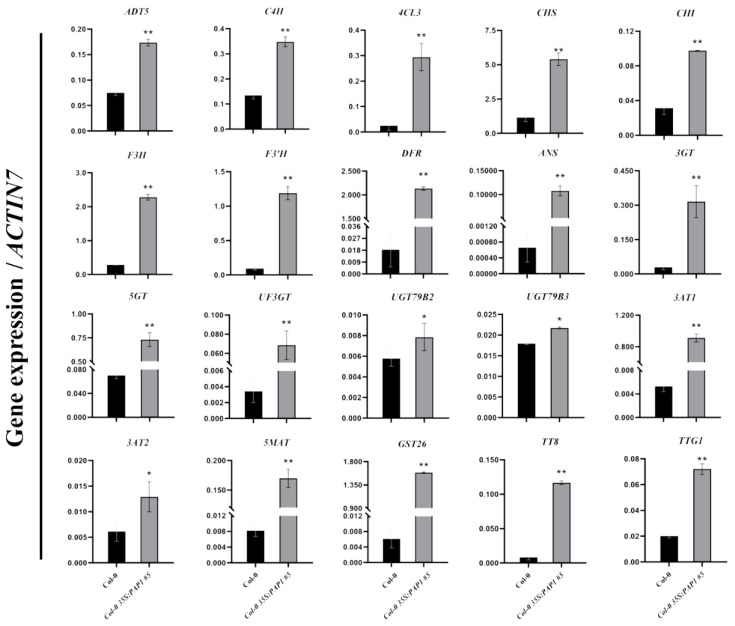
RT-qPCR analysis of the expression of anthocyanin biosynthesis-related genes in the wild-type (Col-0) and *Col-0 35S:PAP1 #5* developing seeds at 12 DAP. Results were normalized against the expression of *Arabidopsis ACTIN7* as the internal control. Values are means ± SD (*n* = 3). ** *p* ≤ 0.01 and * *p* ≤ 0.05 indicate highly significant or significant differences in gene expression levels between wild-type (Col-0) and *Col-0 35S:PAP1 #5* plants (two-tailed paired Student’s *t*-test).

**Figure 4 ijms-24-16049-f004:**
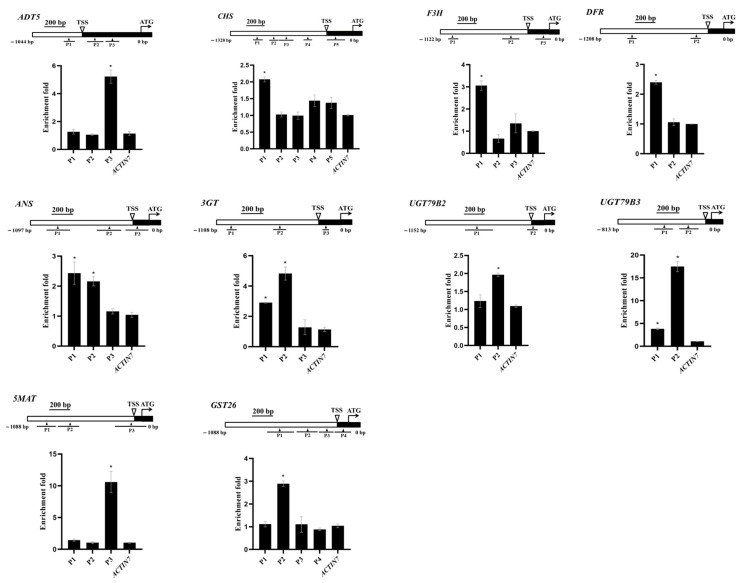
PAP1 targets *ADT5*, *CHS*, *F3H*, *DFR*, *ANS*, *3GT*, *UGT79B2*, *UGT79B3*, *5MAT*, and *GST26* promoters and directly promotes their expressions in developing *Arabidopsis* seeds. Schematic diagrams show the promoter regions of *ADT5*, *CHS*, *F3H*, *DFR*, *ANS*, *3GT*, *UGT79B2*, *UGT79B3*, *5MAT*, and *GST26*, while ChIP-qPCR assays show PAP1 binding to their promoter regions in the developing *Arabidopsis* siliques at 12 DAP. The transcriptional start site (TSS) and exons are represented by black boxes, whereas promoter regions are represented by white boxes. The triangle represents the MYB-recognizing element (MRE) site ANCNNCC and the black lines represented the DNA fragments amplified in ChIP assays for each gene. The enrichment fold of each fragment was calculated first by normalizing the amount of a target DNA fragment against a genomic fragment of *Arabidopsis EF1aA4* as the internal control. Then, the value for *Col-0 35S:PAP1 #5* was normalized against it for wild-type (Col-0) plants. The *Arabidopsis ACTIN7* fragment was amplified as the negative control. Values are means ± SD (*n* = 3). Significant differences in comparison with the *ACTIN7* fragment enrichment are indicated with asterisks (*) (two-tailed paired Student’s *t*-test, *p* ≤ 0.05).

**Figure 5 ijms-24-16049-f005:**
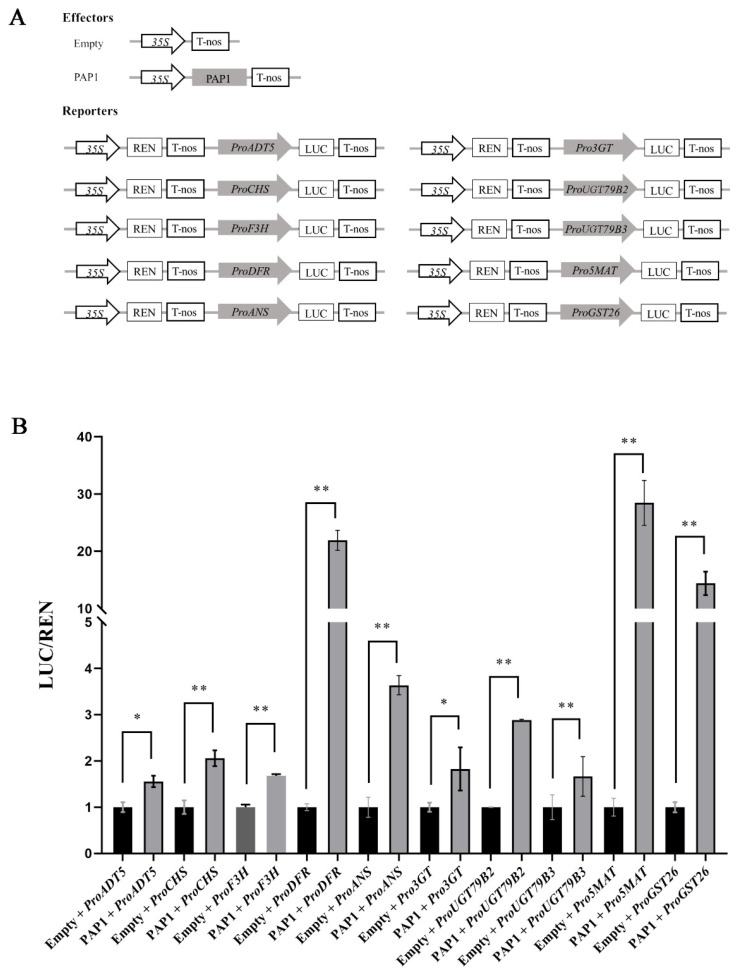
PAP1 directly activates *ADT5*, *CHS*, *F3H*, *DFR*, *ANS*, *3GT*, *UGT79B2*, *UGT79B3*, *5MAT*, and *GST26* transcription in *N. benthamiana* leaves: (**A**) Schematic diagrams show the effectors with and without PAP1 and the reporters containing *ADT5*, *CHS*, *F3H*, *DFR*, *ANS*, *3GT*, *UGT79B2*, *UGT79B3*, *5MAT*, and *GST26* promoters. (**B**) Transient dual-luciferase reporter assay. The reporter constructs were transiently expressed in *N. benthamiana* leaf cells together with empty or PAP1 effector constructs. The expression level of *Renilla* (*REN*) was used as an internal control, and the LUC/REN represents the relative activity of *ADT5*, *CHS*, *F3H*, *DFR*, *ANS*, *3GT*, *UGT79B2*, *UGT79B3*, *5MAT*, and *GST26* promoters. Values are means ± SD (*n* = 6). Double asterisks (**) and asterisk (*) separately indicate highly significant (*p* ≤ 0.01) and significant differences (*p* ≤ 0.05), respectively, in the LUC/REN compared to PAP1 effector with an empty effector (two-tailed paired Student’s *t*-test).

**Figure 6 ijms-24-16049-f006:**
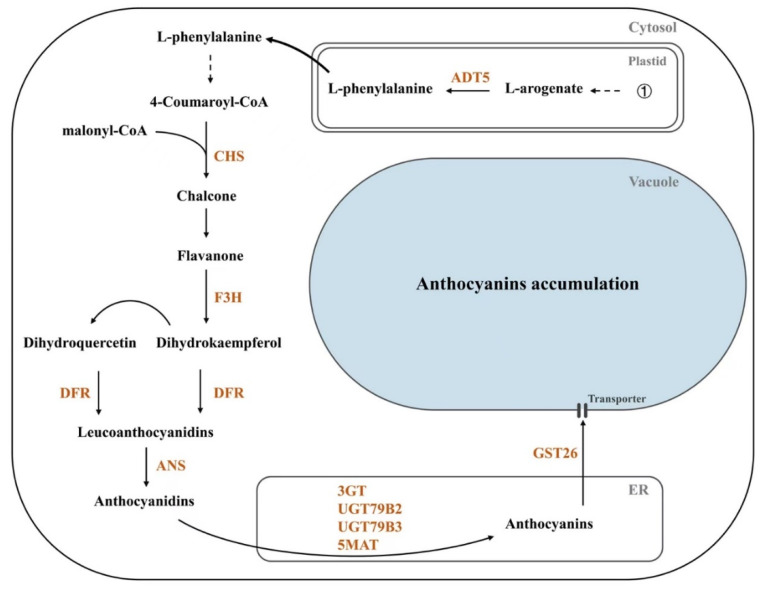
A simplified scheme shows that PAP1 directly regulates the expression of structural genes that control the accumulation of seed anthocyanin in *Arabidopsis*. ①: Shikimic pathway; ADT: arogenate dehydratase; CHS: chalcone synthase; F3H: flavanone 3-hydroxylase; DFR: dihydroflavonol-4-reducatse; ANS: anthocyanidin synthase; 3GT: flavonoid 3-*O*-glycosyltransferase; UGTs: UDP-glycosyltransferases; 5MAT: anthocyanidin 5-*O*-glucoside-6″-*O*-malonyltransferase; GST: glutathione *S*-transferase; ER: endoplasmic reticulum.

**Table 1 ijms-24-16049-t001:** Functional classification of differentially expressed genes (DEGs) in developing seeds between wild-type Col-0 and *Col-0 35S:PAP1 #5* plants at 12 days after pollination (DAP).

Category	Up-Regulated DEGs				Down-Regulated DEGs			
	≥2	1 to 2	Total	Percentage	≤−2	−2 to −1	Total	Percentage
	*log*_2_ *ratio*				*log*_2_ *ratio*			
Metabolism								
Photosynthesis	35	73	108	2.3	1	1	2	0.5
Cell wall	74	81	155	3.3	1	3	4	1.0
Flavonoid metabolism	43	31	74	1.6	1	2	3	0.7
Carbohydrate metabolism	225	382	607	12.8	18	21	39	9.4
Nucleic acid	92	155	247	5.2	12	10	22	5.3
Amino acid and protein	83	156	239	5.0	6	11	17	4.1
Growth and development								
Leaf and root development	17	31	48	1.0	2	5	7	1.7
Shoot development	3	14	17	0.4	0	2	2	0.5
Embryo/seed development	19	42	61	1.3	5	5	10	2.4
Flower development	38	46	84	1.8	3	6	9	2.2
Cell growth	46	68	114	2.4	3	5	8	1.9
Hormone	25	27	52	1.1	2	5	7	1.7
Stress/defense response	315	465	780	16.4	47	48	95	22.9
Cell regulation								
Transcriptional regulation	46	86	132	2.8	5	5	10	2.4
Signaling transduction	50	84	134	2.8	3	7	10	2.4
Transport facilitation	92	172	264	5.5	14	15	29	7.0
Others	695	949	1644	34.5	55	85	140	33.8

Note: Percentage refers to the ratio of genes of each functional category relative to total upregulated or downregulated DEGs identified in the RNA-seq experiment. The DEGs with log_2_ ratios greater than 1 or less than −1 (only Gene Ontology Slim identifiers with *p* ≤ 0.05 and FDR ≤ 0.05) are listed.

**Table 2 ijms-24-16049-t002:** Differentially expressed genes (DEGs) contributing to anthocyanin biosynthesis in the developing seeds between wild-type Col-0 and *Col-0 35S:PAP1 #5* plants at 12 days after pollination (DAP).

Gene Name	ID	Log_2_ Ratios (*Col-0 35S:PAP1 #5*/Col-0)	Functions
*ADT5*	At5g22630	2.47	promoting anthocyanin accumulation [38]
*C4H*	At2g30490	1.83	promoting anthocyanin accumulation [39,40]
*4CL3*	At1g65060	5.43	promoting anthocyanin accumulation [41]
*CHS*	At5g13930	2.26	promoting anthocyanin accumulation [42,43,44,45]
*CHI*	At3g55120	3.08	promoting anthocyanin accumulation [45,46]
*F3H*	At3g51240	2.92	promoting anthocyanin accumulation [47]
*F3’H*	At5g07990	4.06	promoting anthocyanin accumulation [48,49]
*DFR*	At5g42800	7.32	promoting anthocyanin accumulation [42,50]
*ANS*	At4g22880	8.54	promoting anthocyanin accumulation [50,51,52,53,54]
*3GT*	At5g17050	2.62	anthocyanin modification [28]
*5GT*	At4g14090	8.99	anthocyanin modification [28]
*UF3GT*	At5g54060	10.71	anthocyanin modification [55]
*UGT79B2*	AT4G27560	1.36	anthocyanin modification [5]
*UGT79B3*	AT4G27570	3.35	anthocyanin modification [5]
*3AT1*	At1g03940	6.65	anthocyanin modification [56]
*3AT2*	At1g03495	5.09	anthocyanin modification [56]
*5MAT*	At3g29590	11.64	anthocyanin modification [56,57]
*GST26*	At5g17220	9.36	anthocyanin transport [58]
*TT8*	AT4G09820	3.92	promoting anthocyanin accumulation [59,60]
*TTG1*	AT5G24520	0.96	promoting anthocyanin accumulation [61,62]

Note: DEGs with |log_2_ ratios| ≥ 1, and only Gene Ontology Slim identifications with a false discovery rate ≤ 0.05, are listed.

## Data Availability

Data are contained within the article and Appendix A.

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
