# Peer review of "Genome-Wide Identification of PAP1 Direct Targets in Regulating Seed Anthocyanin Biosynthesis in Arabidopsis"

_ijms, 2023, doi:10.3390/ijms242216049_

Round 1

Reviewer 1 Report

Comments and Suggestions for Authors

The paper brings a contribution to the understanding of PAP1's role in regulating anthocyanin biosynthesis in developing seeds of Arabidopsis, aiming to identify PAP1 direct targets in regulating seed anthocyanin biosynthesis; it effectively employs a range of experimental methods and presents comprehensive results. The rationale for the study is well-supported, addressing a gap in previous research that primarily focused on seedlings rather than seeds. The research employs a variety of experimental methods, including RNA-Sequencing analysis, quantitative real-time PCR, chromatin immunoprecipitation assay and dual luciferase reporter assay; this multi-pronged approach enhances the robustness of the results, which are presented in a detailed and organized manner, accompanied by figures and tables that effectively illustrate key findings. The inclusion of phenotypic data, gene expression patterns and functional classifications enhances the comprehensibility of the results. The manuscript successfully integrates existing knowledge about the role of PAP1 and the MYB-bHLH-WD40 protein complex in anthocyanin biosynthesis; this context provides a foundation for the new findings and their implications.

While the paper is generally well-written, there is still room for improvement, hence consider the following issues:

L.194 - "extremely significant" should be replaced with "highly significant" or similar.

L.423, 468 – add type and producer for the centrifuge;

L.426 – add type and producer for the spectrophotometer;

L.467 – add type and producer for the ultrasonic bath;

L.474 – replace Ph > pH;

L.4907 – add type and producer for the climate incubator;

- the paper could elaborate more on the biological relevance of the findings, for instance, how might the identified targets influence seed development or respond to environmental stimuli? (as suggested in L.34)

Comments on the Quality of English Language

Only minor editing of English language are required

Author Response

Dear Reviewer,

We have taken all comments from you into consideration, and made a thorough revision on this manuscript. Please see the point-by-point response to you in attachment file.

We thank you again for the suggestions which significantly improved this manuscript. We deeply appreciate your effort and consideration!

Sincerely,

Mingxun Chen

Reviewer 2 Report

Comments and Suggestions for Authors

Article does not flow well on first reading.

They should show in more detail the involvement of PAP 1 in the various regulatory pathways related to the anthocyanin pathway to better understand the introduction.

I don't have additional figures and tables.

Results explained in a concise manner that could be expanded to improve clarity. The figures and tables are good and illustrative.

Excellent discussion to explain the results of the experiment.

Numerous materials and methods described in detail.

Innovative study regarding Arabidopsis seeds compared to previous approaches.

Comments on the Quality of English Language

Moderate English changes required

Author Response

(The authors gave the same response as above.)
